# Openness of Fish Habitat Matters: Lake Pelagic Fish Community Starts Very Close to the Shore

Karlos Moraes [1,2,†], Allan T. Souza [1,†], Mojmír Vašek [1,†], Daniel Bartoň [1], Petr Blabolil [1], Martin Čech [1], Romulo A. dos Santos [1], Vladislav Draštík [1], Michaela Holubová [1], Tomáš Jůza [1], Luboš Kočvara [1], Kateřina Kolářová [1,3], Josef Matěna [1,‡], Jiří Peterka [1], Milan Říha [1], Zuzana Sajdlová [1], Marek Šmejkal [1], Lobsang Tsering [1] and Jan Kubečka [1,2,*]

1    Institute of Hydrobiology, Biology Centre of the Czech Academy of Sciences, Na Sádkách 7, 370 05 České Budějovice, Czech Republic; karlos.moraes@hbu.cas.cz (K.M.); Allan.Souza@hbu.cas.cz (A.T.S.); mojmir.vasek@hbu.cas.cz (M.V.); daniel.barton@hbu.cas.cz (D.B.); petr.blabolil@hbu.cas.cz (P.B.); carcharhinusleucas@yahoo.com (M.Č.); romulo.ufsc@gmail.com (R.A.d.S.); vladislav.drastik@hbu.cas.cz (V.D.); michaela.holubova@hbu.cas.cz (M.H.); tomas.juza@hbu.cas.cz (T.J.); lubos.kocvara@hbu.cas.cz (L.K.); Katerik@seznam.cz (K.K.); jiri.peterka@hbu.cas.cz (J.P.); riha.milan@centrum.cz (M.Ř.); zuzana.sajdlova@bc.cas.cz (Z.S.); mareks1@centrum.cz (M.Š.); lobsang.tsering@hbu.cas.cz (L.T.)
2    Faculty of Science, University of South Bohemia, Branišovská 31, 370 05 České Budějovice, Czech Republic
3    Vltava River Authority, E. Pittera 1622/1, 370 01 České Budějovice, Czech Republic
*    Correspondence: kubecka@hbu.cas.cz
†    These three authors contributed equally to this work.
‡    This paper is dedicated to the living memory of Josef Matěna who deceased during preparation of this paper in September 2021.

**Abstract:** Fish communities differ significantly between the littoral and the pelagic habitats. This paper attempts to define the shift in communities between the two habitats based on the European standard gillnet catch. We sampled the benthic and pelagic habitats from shore to shore in Lake Most and Římov Reservoir (Czech Republic). The 3 m deep pelagic nets were spanned across the water body at equal distances from two boundary points, where the depth was 3.5 m. The benthic community contained more fish, more species, and smaller individuals. The mild sloped littoral with a soft bottom attracted more fish than the sloping bank with a hard bottom and less benthos and large *Daphnia*. The catch of the pelagic nets was dominated by eurytopic fish—rudd (*Scardinius erythrophthalmus*) and roach (*Rutilus rutilus*) in Most and bleak (*Alburnus alburnus*) in Římov. With the exception of one case where overgrown macrophytes extended the structured habitat, the largest shift from the benthic to the pelagic community was observed only in the first pelagic gillnet above the bottom depth of 3.5 m. Open water catches were relatively constant with small signs of decline towards the middle of the lake. The results indicate that the benthic gillnet catch is representative of a very limited area and volume, while most of the volume is dominated by the pelagic community. This has important consequences for the assessment of the community parameters of the whole lake following the European standards for gillnet sampling design.

**Keywords:** habitat use; spatial distribution; ecotone; open water *Scardinius*; *Rutilus*; *Alburnus*; *Perca*

## 1. Introduction

In any environment, species composition changes gradually or abruptly between habitats. These ecological gradients have been the subject of a number of studies in ecology and usually reflect the abundance and richness of species [1–4]. Rapid changes in species ecological gradients, termed ecotones, have been observed in a variety of ecosystems [1]. Ecotones can affect the abundance and distribution of organisms.

The distribution of fish species is not random and their distribution in different habitats depends on several factors, including substrate composition [5–7], depth [8–10], habitat

complexity [5,9,11,12], temperature [13–15], oxygen concentration [13,16], distribution of planktonic and benthic organisms [14,17,18], and other factors. Biotic and abiotic factors both influence fish distribution, but the contribution of each variable is difficult to separate, especially because of seasonal variance [19]. We can assume that the spatial distribution of fish is optimized by strategies to maximize habitat and resource use with the aim of increasing individual fitness [20]. Predation risk must also be considered. In a dynamic environment, defining habitat boundaries for highly vagrant species is a challenging task. In lentic ecosystems, the boundary between the littoral (shallow areas) and pelagic zones is poorly understood [21].

The definition of the pelagic and littoral zones is based on the physical or biological characteristics of lentic ecosystems. Pelagic zones, also referred to as open water, are usually assumed to be the deeper areas of water bodies characterized by the absence of bottom or habitat structure [8]. Primary production in the pelagic zone is highly dependent on phytoplankton and is therefore often lower than in the littoral zones [22]. On the other hand, littoral zones are usually defined as nearshore areas where light intensity is sufficient to reach the bottom sediment and allow the primary producers (macrophytes and algae) to thrive [23].

For prey fish, open water is far less safe than the structured littoral. Fernando and Holčík [24] proposed a theory that fishes in evolutionarily young Palearctic systems are mostly of riverine origin and are not sufficiently specialized to take advantage of the pelagic production. Consequently, they expected far fewer fish in the pelagic regions. This is especially true for small fish during the day. Dense schools of species such as juvenile perch, *Perca fluviatilis*; bleak, *Alburnus alburnus*; roach, *Rutilus rutilus*; asp, *Leuciscus aspius*; and dace, *Leuciscus leuciscus,* reside in the littoral during the day [25]. The productivity of the spatially restricted littoral is often insufficient to maintain such high fish densities, forcing even small fish to migrate to offshore during the dusk to feed and return to the littoral before dawn [25–27]. Larger fish, which are less threatened by predators, stay in open water during the day and partially migrate ashore in the evening [28–30]. These patterns have been formulated based on active sampling techniques and hydroacoustics. With the tremendous expansion of the European Standard gillnets (ESG), which are currently the most common sampling tool on the continent, sampling thousands of lakes [31,32], the reflection of the above patterns in gillnet catches becomes relevant. The ESG catches include evening, night, and morning events, and particularly reflect fish abundance and activity. It is well known that the benthic fish community in ESG catches differs from the pelagic one [33–35]. This is true even in smaller waters [8,11]. However, we are not aware of any study showing exactly where the benthic/littoral community transitions to the pelagic and how gradual or abrupt this transition is. Identifying the boundaries between the benthic and pelagic zones is important for estimating fish catch per unit effort [33], or other characteristics of fish from across the lake such as size or age distribution [36]. To date, only arbitrarily assumed values such as distance from the bottom 1.5 m [31,36] or 3 m [33], or even 1.5–3 m depending on the thickness of the layer [35], have been used without verification to draw the boundary between the two habitats.

The aim of this study is to investigate the proportions of the littoral and the pelagic communities in two water bodies with different environmental conditions (Most Lake and Římov Reservoir). With regard to the preference of the littoral over the pelagic, we can divide the fish community into three possible groups: the most diverse benthic fishes, which are strongly bound to the bottom and the littoral zone, the eurytopic fishes, which can use both the littoral and the pelagic habitat, and the open-water fishes, which prefer the pelagic habitat in open waters. Our hypothesis is that during the transition from the littoral to the pelagic, the benthic-bound species will greatly decrease, while the open-water species will greatly increase, and the eurytopic species will show little variation.

## 2. Materials and Methods

### 2.1. Sampling Sites

Two water bodies in the Czech Republic, Lake Most and the Římov Reservoir, were chosen for the experiment (Figure 1). Most is a post-coal mining lake (Ústí nad Labem region, 50.54 N, 13.65 E, see Figure 1) with an area of 310 ha, a maximum depth of 75 m, and a mean depth of 22 m. The lake was formed after the termination of coal mining in the summer of 1999 due to the filling of the open pit from autumn 2008 to autumn 2012. Most is an oligotrophic lake with a high abundance of macrophytes in its littoral area and a high water transparency [37]. paracol

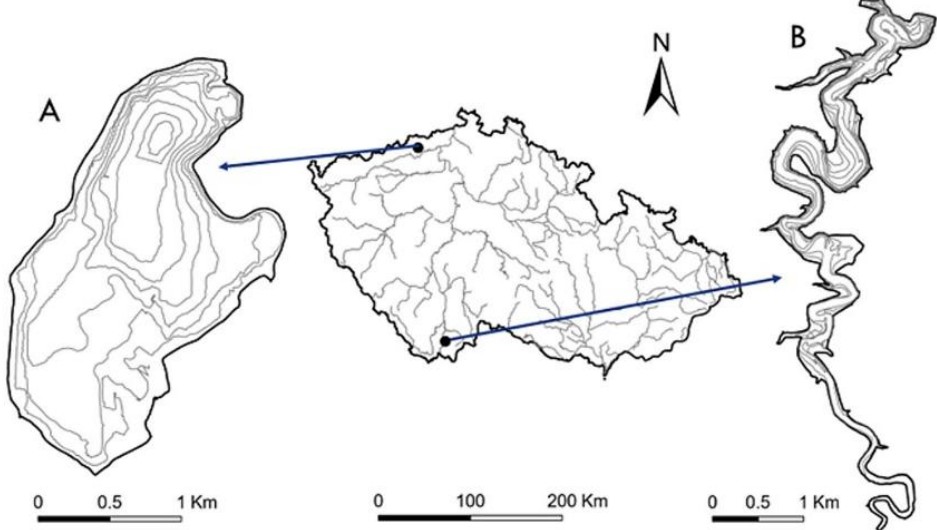

**Figure 1.** Map of Czech Republic with major rivers and the location and shape of Most Lake (**A**) and Římov Reservoir (**B**) with their respective bathymetries.

The Římov Reservoir (South Bohemia Region, 48.848 N, 14.487 E) is a canyon-shaped reservoir with a narrow (max. width 600 m) and elongated shape (length 10 km). The reservoir was built during the 1970s (from 1971 to 1978) and covers an area of about 200 ha with a volume of $34.3 \times 10^6$ m$^3$, a maximum depth of 40 m, and an average depth of 12 m. Compared to the Most Lake, the Římov Reservoir has a gently to steeply sloping shore without submerged vegetation, which is missing due to significant water level fluctuations and low water transparency due to the eutrophic status of the water [38].

### 2.2. Gillnet Sampling in General

The European Standard gillnets (ESG) [39] were used to estimate the association of fish with littoral and pelagic habitats. The benthic ESG gillnet with 1.5 m height × 30 m length and 2.5 m mesh panels for each of the 12 mesh sizes was deployed in the littoral, while the pelagic gillnet with 3 m height × 30 m length and 2.5 m mesh panels for each of the 12 mesh sizes was deployed in the open water. The mesh sizes of the ESG followed a geometric series with a ratio of approximately 1.25 (5, 6.25, 8, 10, 12.5, 15.5, 19.5, 24, 29, 35, 43, and 55 mm) in random order. The first pelagic gill net from the shore was deployed above the bottom depth of 3.5 m (bottom line of the gillnet 0.5 m above the bottom). Depth was measured using a Humminbird Piranha echo sounder operating at 200 kHz.

Gillnet deployment was from bank to bank (Figures 2 and 3). The opposite banks differed in bottom slope, as is common in riverine waterbodies, and the fish community is influenced by the slope [40]. The mild sites had a bank slope of less than 8°, while the steep sites had a slope of more than 15°. The pelagic nets were laid out equidistantly, from the first pelagic net of the mild side to the first pelagic net of the steep side (Figures 2 and 3). The gillnets were named according to the slope of the bank on which they were deployed.

The benthic gillnets were named MB (mild benthic—benthic net on the mild slope) and SB (steep benthic), while the pelagic gillnets were named MP (mild pelagic—pelagic net on the side, adjacent to the mild slope) and SP (steep pelagic), with one pelagic gillnet deployed in the center of the lake (mid distance between the two 3.5 m isobaths in the sampled area) referred to as the center net. Given that the pelagic area had more nets, the number immediately following the acronym indicates the number of gillnets deployed from the shore, e.g., SP1 is the first pelagic gillnet from the steep shore, while SP2 is the following pelagic gillnet, and so on (Figures 2 and 3).

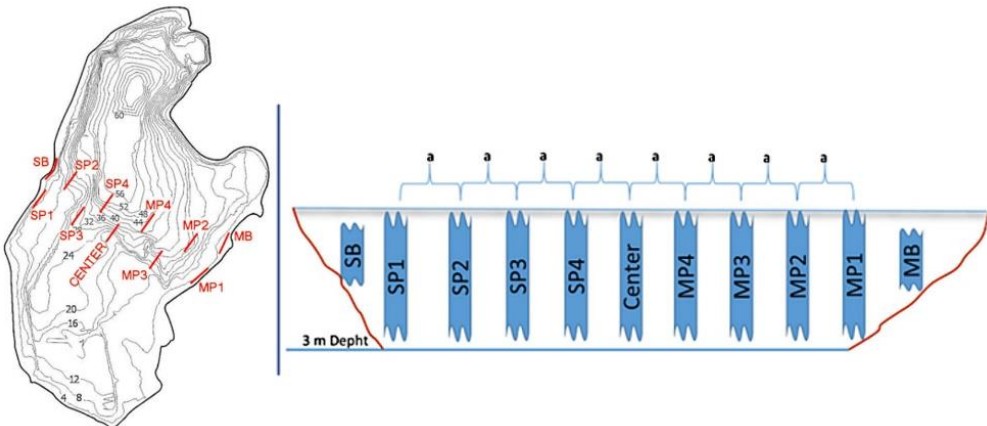

**Figure 2.** Sampling sites in the shape of Most Lake, on the left, and gillnet deployment scheme, on the right. SB—benthic gillnet at steep slope; SP1—center—MP1—pelagic gillnets; MB—benthic gillnet at mild slope. See Material and Methods for detailed explanations.

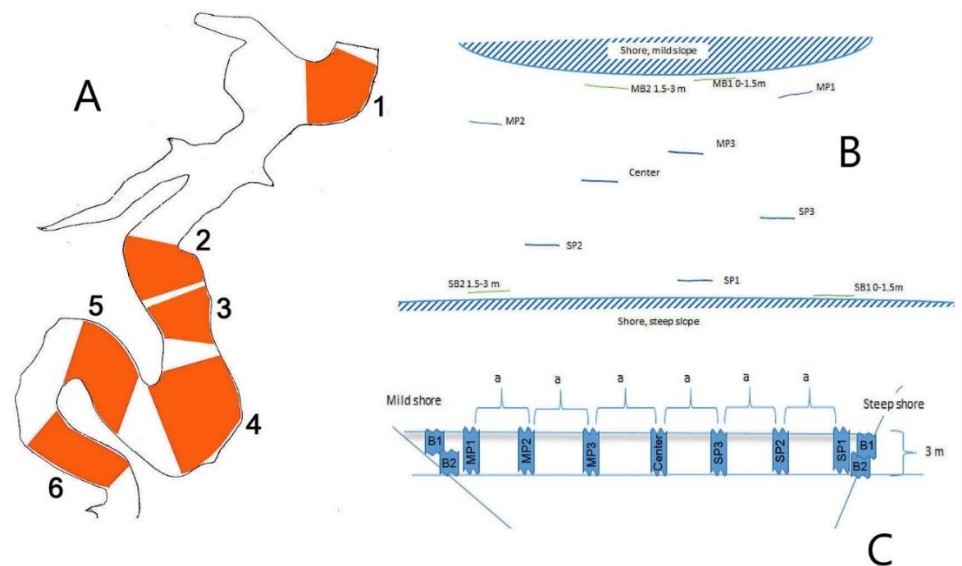

**Figure 3.** Sampling sites (orange markings in the shape of Římov Reservoir, (**A**), gillnet deployment scheme for each area (**B**) and net scheme from shore to shore (**C**). B1—benthic gillnet at depth 0–1.5 m; B2—benthic gillnet at depth 1.5–3 m; SP1—center—MP1—pelagic gillnets. See Material and Methods for detailed explanations.

Sampling was done in accordance with CEN standards [39], with the gillnets being deployed 2 h before the sunset and lifted 2 h after the sunrise [41]. Standard fish length and weight of all the captured individuals were measured to the nearest mm and g, respectively. The catch per unit of effort (CPUE) was defined as the number of individuals per 1000 m$^2$ of net per night, analogically the sampled biomass per unit of effort (BPUE) was defined as number of grams per 1000 m$^2$ of a net per night.

## 2.3. Most Lake Sampling Design

Sampling in Lake Most was conducted from 3–6 September 2018. The transect area in Lake Most was in the form of a ribbon that extended from shore to shore across the lake. We sampled two benthic and nine pelagic sites in the lake (Figure 2). The east mild shore had a slope of 7° of declination and the west steep shore had a slope of 15°. At each littoral or pelagic location shown in Figure 2, we set up three ESG nets connected by a 40 m rope to ensure adequate spacing between them. The distance between gillnet sampling locations was 150 m. The benthic gillnets were deployed at a depth of 1.5–2.5 m, and the first pelagic gillnet was deployed at a depth greater than 3.5 m from each bank. The remaining pelagic gillnets were deployed at the same spacing. The gillnets were deployed parallel to the shore. Altogether 6 benthic and 27 pelagic gillnets were deployed.

## 2.4. Římov Reservoir Sampling Design

Sampling of the Římov Reservoir was conducted from 30 July to 2 August 2019. Six locations in the reservoir were sampled, each with both a mild slope shore (2° to 8° slope) and a steeply sloping bank (20° to 35° slope). We selected sites only in the true lacustrine zone (Figure 3) to avoid the change in productivity that increases further upstream [42]. A single ESG device was deployed at each net location of each site. The nets were scattered to ensure that no net interfered with the others (Figure 3B). The minimum distance between nets was 60 m, but usually it was more than 100 m. For this experiment, we also deployed two sets of benthic gillnets on either side of the reservoir (one in the 0–1.5 m depth range and the second in 1.5–3 m). For this article, the CPUE and BPUE values from these two nets were combined so that they well represent the littoral range of 0–3 m. Altogether, 24 benthic and 42 pelagic gillnets were deployed.

Zooplankton samples were collected 30–60 min after each gillnet deployment. Vertical hauls with a plankton net (diameter 20 cm, mesh size 0.2 mm) were made at both ends of each pelagic net. Hauls were made from 3 m depth to the surface and two hauls were combined in each zooplankton sample. Samples of littoral zooplankton were collected using a Schindler sampler (volume 30 L, mesh size 0.2 mm). Each sample of littoral zooplankton was collected by combining two samples (one from the upper, 0–1.5 m, and one from the lower, 1.5–3 m, portion of the sampled layer up to 3 m) in one bottle. Samples of littoral zooplankton were collected from both ends of the benthic gillnets deployed in the littoral zones. The zooplankton was divided into 3 groups: *Daphnia galeata*, other Cladocera (*Acroperus harpae*, *Bosmina coregoni*, *Bosmina longirostris*, *Ceriodaphnia quadrangula*, *Diaphanosoma brachyurum*, *Chydorus sphaericus*, *Leptodora kindti*, and *Leydigia leydigi*), and Copepoda (*Cyclops vicinus*, *Eudiaptomus gracilis*, *Mesocyclops leuckarti*, *Thermocyclops crassus*, *Thermocyclops oithonoides*, Cyclopoida–copepodites, and Diaptomidae-copepodites). In addition, 100 individuals of *D. galeata* were measured for body size in each zooplankton sample. The body size of the *Daphnia* was measured from the top of the head to the base of the caudal spine. An amount of 1 mm of the body length of the *Daphnia* was chosen as the threshold between the small and the large individuals.

Benthic samples were collected from the same six locations where benthic gillnets were set. We conducted kick sampling [43] at two depths, 0.3 m and 1 m, for 2 min using a bar net with a mesh size of 0.4 mm. We also attempted to sample the 2 m depth with a 20 cm × 20 cm Eckmann grab, but sampling was often unsuccessful due to the hard substrate, especially on steep banks. The benthic macroinvertebrates were divided into 4 groups: Permanent fauna (*Hydra*, *Planaria*, *Stylaria*, *Nais*, *Dero*, Tubificidae, Nematoda, *Helobdella*, *Lymnaea*, and *Asellus*), Ephemeropteran larvae (*Caenis*, *Cloeon*, and *Ephemera*), Chironomid larvae (*Ablabesmyia*, *Corynoneura*, *Cricotopus*, *Cryptochironomus*, *Dicrotendipes*, *Polypedilum*, *Glyptotendipes*, *Endochironomus*, and *Tanytarsus*); and the other temporal fauna (Zygoptera, Leptoceridae, *Ecnomus*, Limnephilidae, Tabanidae, *Sialis*, Hydrophilidae, and *Micronecta*).

*2.5. Data Analyses*

Catch per unit effort (CPUE) was calculated as the mean of the total number of individuals divided by the total sampling effort (net surface area), while biomass per unit effort (BPUE) was calculated as the total weight of catch per 1000 m$^2$ of net area. CPUE and BPUE were calculated for individual species as well as for the entire fish assemblage.

Negative binomial generalized linear models (GLM) were applied to describe the differences in fish CPUE and BPUE values (CPUE and BPUE) with distance from shore in Most. The negative binomial generalized linear model was chosen because it can cope with a large number of zeros and over-dispersed data [44]. The MASS package was used to compute all GLMs [45].

For the analyses in the Římov Reservoir, a generalized linear mixed effects model (GLMM) fitted for the negative binomial family was used, with localities included in the model as a random effect. The model was applied to describe differences in fish CPUE and BPUE and zooplankton density as a function of distance from shore, as well as benthic macroinvertebrate numbers on gentle and steep slopes and at different depths. All data analyses were performed using R software [46].

Diversity indices (Shannon–Weaver, Simpson, Pielou's evenness, and richness) of fish communities were also calculated using the Vegan package of the R software [47].

## 3. Results

*3.1. Most*

A total of 881 fishes belonging to five different species were captured. The most abundant species were the roach (*Rutilus rutilus*)—62.43%; rudd (*Scardinius erythrophthalmus*)—29.63%; European perch (*Perca fluviatilis*)—7.72%; northern pike (*Esox lucius*) and ruffe (*Gymnocephalus cernua*), both with 0.11% of the total fish captured.

CPUE values for the entire fish community decreased sharply from the shore to the center of the lake, on both mild ($p < 0.001$, deviance = 18.6) and steep ($p < 0.001$, deviance = 15.5) shores (Figure 4). This pattern is particularly striking from the first pelagic gill net SP1 to the middle gill net, which had the lowest CPUE values among all gill nets used (Table 1).

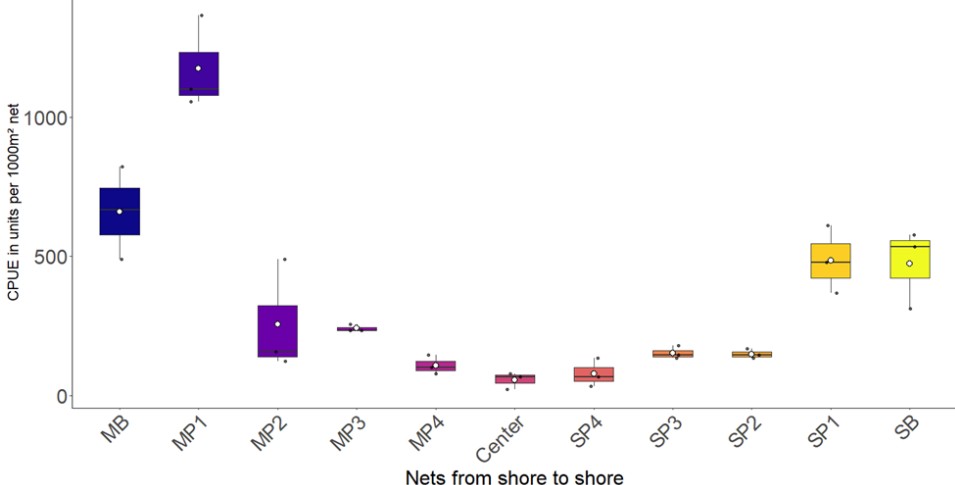

**Figure 4.** Total catch per unit effort (CPUE; individuals per 1000 m$^2$ of net) from 11 gillnets fished in Most Lake. The boxplot represents the quartile value of the CPUE, the grey dots represent the CPUE of three individual nets deployed at the same distance from shore, the thick middle line represents the median, and the white dot represents the arithmetic mean. The site MP1 was surrounded by overgrown macrophytes.

**Table 1.** Catch per unit of effort (inds. 1000 m$^{-2}$ of gillnets), standard errors (*se*), and the significance level (*p*) at various benthic and pelagic sites of Most Lake. See Section 2 for detailed explanations of gillnet locations.

| Species | MB | se | MP1 | se | MP2 | se | MP3 | se | MP4 | se | Center | se | p_Value |
|---|---|---|---|---|---|---|---|---|---|---|---|---|---|
| *Esox lucius* | 0 | 0 | 0 | 0 | 0 | 0 | 0 | 0 | 3.70 | 3.70 | 0 | 0 | ns |
| *Gymnocephalus cernua* | 0 | 0 | 0 | 0 | 0 | 0 | 0 | 0 | 0 | 0 | 0 | 0 | ns |
| *Perca fluviatilis* | 177.78 | 22.22 | 29.63 | 9.80 | 7.41 | 0 | 0 | 0 | 0 | 0 | 0 | 0 | ns |
| *Rutilus rutilus* | 400.1 | 111.1 | 859.26 | 53.9 | 181.5 | 94.57 | 137.04 | 9.80 | 55.56 | 12.80 | 18.52 | 13.35 | 0.00 |
| *Scardinius erythrophthalmus* | 81.48 | 45.07 | 285.19 | 53.4 | 66.67 | 27.96 | 103.7 | 3.70 | 48.15 | 9.80 | 37.04 | 9.80 | 0.00 |
| **Species** | **SB** | *se* | **SP1** | *se* | **SP2** | *se* | **SP3** | *se* | **SP4** | *se* | **Center** | *se* | **p_Value** |
| *Esox lucius* | 0 | 0 | 0 | 0 | 0 | 0 | 0 | 0 | 0 | 0 | 0 | 0 | ns |
| *Gymnocephalus cernua* | 7.41 | 0 | 0 | 0 | 0 | 0 | 0 | 0 | 0 | 0 | 0 | 0 | ns |
| *Perca fluviatilis* | 222.22 | 102.64 | 11.11 | 0 | 0 | 0 | 0 | 0 | 3.70 | 0 | 0 | 0 | ns |
| *Rutilus rutilus* | 162.96 | 51.85 | 325.93 | 45.1 | 51.85 | 9.80 | 111.1 | 0 | 14.81 | 7.41 | 18.52 | 13.4 | 0.00 |
| *Scardinius erythrophthalmus* | 81.48 | 60.63 | 148.15 | 16.14 | 96.3 | 18.52 | 40.74 | 13.35 | 59.26 | 20.62 | 37.04 | 9.80 | ns |

At the site of MP1 (bottom depth = 3.5 m), many high macrophyte stands were still present, so the habitat cannot be considered truly pelagic. This may have been the cause of the higher CPUE values at this site (Figure 4). When analyzing each species independently, two of the five species showed significant non-random distribution from bank to bank, namely roach (mild: $p < 0.001$, deviance = 21.9; steep: $p < 0.001$, deviance = 18.3) and rudd (mild: $p = 0.003$, deviance = 21.5).

The BPUE values differed significantly for both the mild slope ($p < 0.001$, deviance = 18.9) and the steep shores ($p < 0.023$, deviance = 16, Figure 5, Table 2). The influence of the macrophyte beds at the site of MP1 was again very evident, with BPUE more than twice that of the other gillnets (Table 2). When analyzing the distribution of the individual species on both slopes, the results followed a similar pattern to the CPUE in the case of roach (mild: $p < 0.001$, deviance = 21.1; steep: $p < 0.002$, deviance = 18.2), with a significant response to distance from both shores, and also perch, but only for the mild slope side (mild: $p < 0.001$, deviance = 12.6).

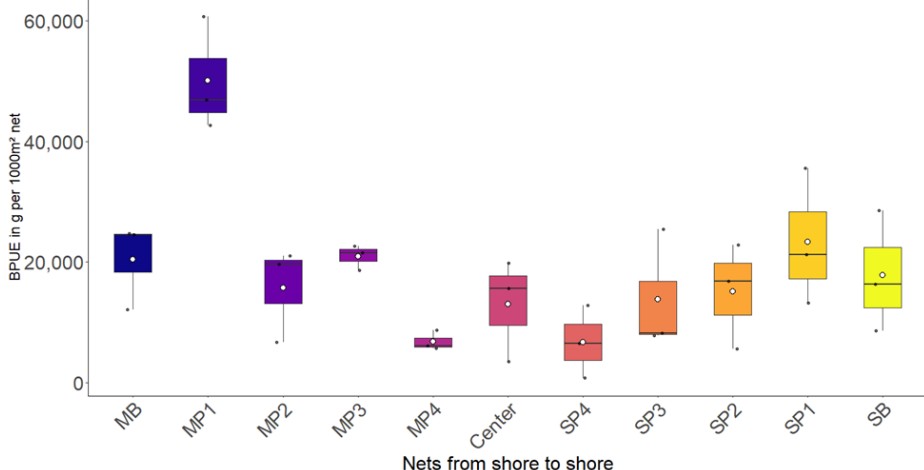

**Figure 5.** Total biomass per unit effort (BPUE; kg per 1000 m$^2$ of net) from 11 gillnets fished in Most Lake. The boxplot represents the quartile value of the BPUE, the grey dots represent the mean of the individual net, the thick middle line represents the median, and the white dot represents the arithmetic mean. The site MP1 was surrounded by overgrown macrophytes.

**Table 2.** Biomass catch per unit of effort (BPUE, g.1000 m$^{-2}$ of gillnets), standard errors (*se*), and the significance level (*p*) at various benthic and pelagic sites of Most Lake. See Section 2 for detailed explanations of gillnet locations.

| Species | MB | se | MP1 | se | MP2 | se | MP3 | se | MP4 | se | Center | se | p_Value |
|---|---|---|---|---|---|---|---|---|---|---|---|---|---|
| *Esox lucius* | 0 | *0* | 0 | *0* | 0 | *0* | 0 | *0* | 607.41 | *0* | 0 | *0* | ns |
| *Gymnocephalus cernua* | 0 | *0* | 0 | *0* | 0 | *0* | 0 | *0* | 0 | *0* | 0 | *0* | - |
| *Perca fluviatilis* | 11,422 | *1824* | 6526 | *3593* | 700 | *0* | 0 | *0* | 0 | *0* | 0 | *0* | 0.0001 |
| *Rutilus rutilus* | 7387 | *3283* | 1611 | *2290* | 8170 | *2073* | 5770 | *763* | 2044 | *742* | 1151 | *1089* | 0.0001 |
| *Scardinius erythrophthalmus* | 1659 | *830* | 27,444 | *8510* | 6911 | *2269* | 15,193 | *1817* | 4207 | *2120* | 11,874 | *4166* | ns |
| **Species** | **SB** | | **SP1** | | **SP2** | | **SP3** | | **SP4** | | **Center** | | **p_Value** |
| *Esox lucius* | 0 | *0* | 0 | *0* | 0 | *0* | 0 | *0* | 0 | *0* | 0 | *0* | - |
| *Gymnocephalus cernua* | 281 | *0* | 0 | *0* | 0 | *0* | 0 | *0* | 0 | *0* | 0 | *0* | ns |
| *Perca fluviatilis* | 12,385 | *4375* | 1233 | *0* | 0 | *0* | 0 | *0* | 955.56 | *0* | 0 | *0* | ns |
| *Rutilus rutilus* | 3911 | *2410* | 6091 | *1774* | 2733 | *1389* | 2503 | *633.28* | 229.63 | *114.99* | 1150 | *1089* | 0.002 |
| *Scardinius erythrophthalmus* | 1259 | *1004* | 16,025 | *3609* | 12,381 | *4658* | 11,348 | *6231* | 5577 | *2549* | 11,874 | *4165* | ns |

Rudd and roach clearly dominated the fish community of Most Lake (Figure 6). They had absolute dominance in the open water habitats, while inshore at the benthic habitat, the dominance was shared with perch (and rarely with ruffe). The dominance of rudd was even more evident in the biomass (Figure 7). This means that, on average, rudd were larger than roach in open waters (see also Table 3). Analysis of the overall size distribution also showed that the lowest mean sizes were found in the first nets on each shore, and the highest mean size would be in the center (*p* < 0.001, deviance = 903.44). Rarely, larger individuals of pike and perch were also caught in the pelagic area (Figure 7). Perch dominated the benthic habitats in terms of biomass, which was significantly different from all pelagic habitats. The diversity indices of the fish community of Lake Most were generally low and showed a weak tendency to decrease towards the center of the lake (Figure 8). The low values correspond to a low number of species present. The presence of littoral elements such as perch and ruffe and the lower dominance of rudd resulted in a slightly higher diversity in the littoral. However, none of the diversity indices showed a significant trend between the littoral and the pelagic.

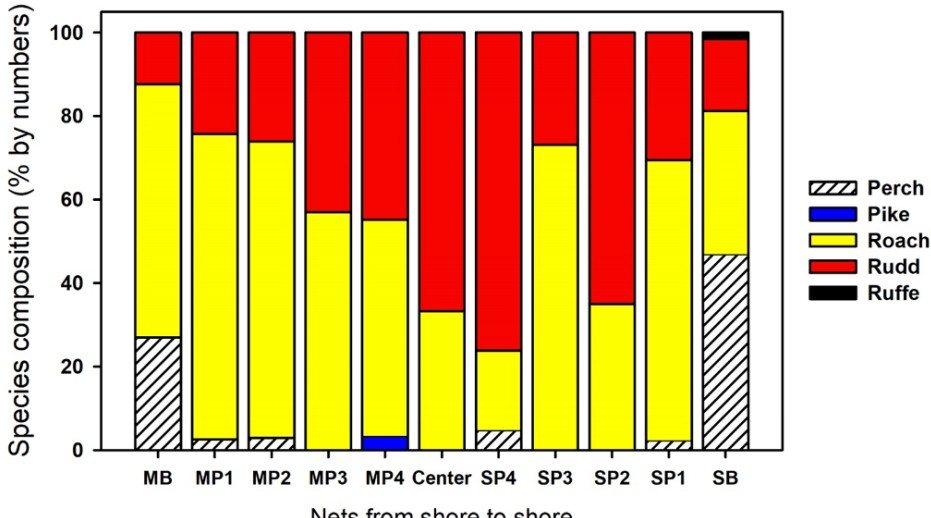

**Figure 6.** Fish species numerical percentual composition at different distances from the shore of Most Lake.

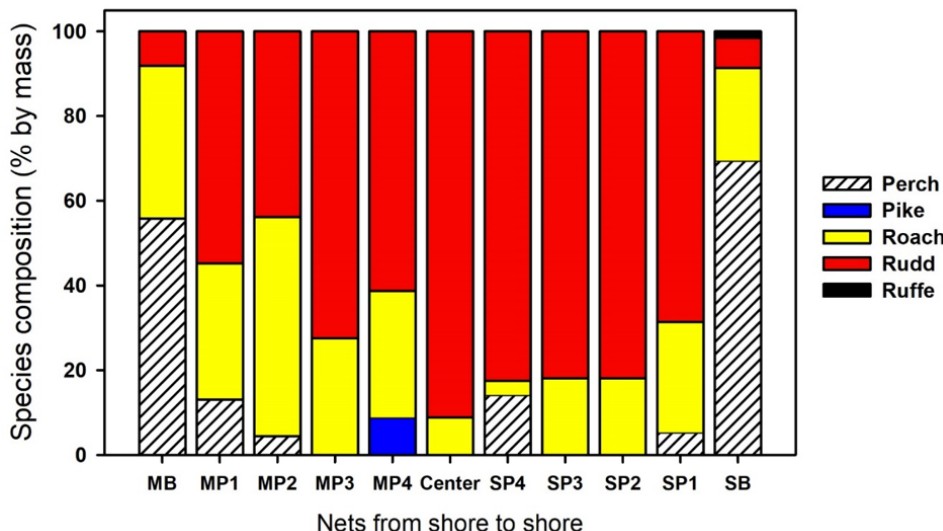

**Figure 7.** Fish species biomass composition at different distances from the shore of Most Lake.

**Table 3.** Average standard length (mm) and standard errors (*se*) of individual species at various benthic and pelagic sites of Most Lake. See Section 2 for detailed explanations of gillnet locations.

| Species | MB | se | MP1 | se | MP2 | se | MP3 | se | MP4 | se | Center | se |
|---|---|---|---|---|---|---|---|---|---|---|---|---|
| *Esox lucius* | 0 | 0 | 0 | 0 | 0 | 0 | 0 | 0 | 260 | 0 | 0 | 0 |
| *Gymnocephalus cernua* | 0 | 0 | 0 | 0 | 0 | 0 | 0 | 0 | 0 | 0 | 0 | 0 |
| *Perca fluviatilis* | 138.33 | 6.87 | 196.25 | 28.53 | 167.5 | 22.5 | 0 | 0 | 0 | 0 | 0 | 0 |
| *Rutilus rutilus* | 85.61 | 2.46 | 89.16 | 1.6 | 109.49 | 6.82 | 108.24 | 7.11 | 110.67 | 9.18 | 126 | 21.35 |
| *Scardinius erythrophthalmus* | 96.91 | 4.79 | 138.25 | 5.81 | 137.22 | 13.64 | 155 | 11.26 | 128.85 | 14.73 | 198.5 | 23.52 |
| **Species** | **SB** | *se* | **SP1** | *se* | **SP2** | *se* | **SP3** | *se* | **SP4** | *se* | **Center** | *se* |
| *Esox lucius* | 0 | 0 | 0 | 0 | 0 | 0 | 0 | 0 | 0 | 0 | 0 | 0 |
| *Gymnocephalus cernua* | 125 | 0 | 0 | 0 | 0 | 0 | 0 | 0 | 0 | 0 | 0 | 0 |
| *Perca fluviatilis* | 131.5 | 6.96 | 161.67 | 32.19 | 0 | 0 | 0 | 0 | 220 | 0 | 0 | 0 |
| *Rutilus rutilus* | 85.14 | 7.63 | 91.89 | 2.41 | 106.07 | 15.35 | 94.5 | 4.39 | 90 | 9.13 | 126 | 21.35 |
| *Scardinius erythrophthalmus* | 86.55 | 4.03 | 145.4 | 8.36 | 136.35 | 13.63 | 195 | 20.83 | 134.69 | 13.58 | 198.5 | 23.52 |

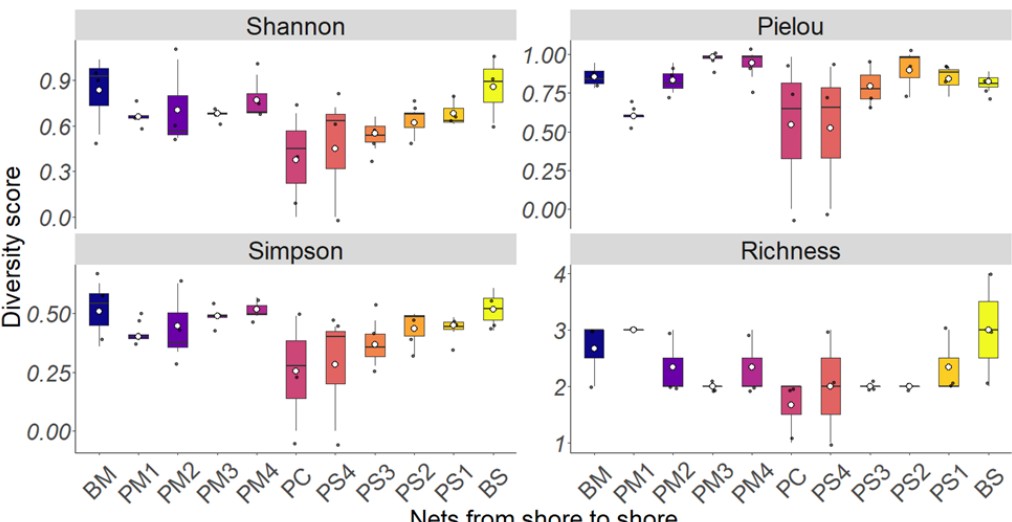

**Figure 8.** Fish diversity indices at different distances from the shore of Most Lake.

### 3.2. Řimov

A total of 5791 fish were caught from nine different species: 76.19% bleak (*Alburnus alburnus*); 13.78% roach; 5.13% perch; 2.19% ruffe; 0.98% asp (*Leuciscus aspius*); 0.98% bream (*Abramis brama*); 0.5% pikeperch (*Sander lucioperca*); 0.14% rudd; and 0.10% wels catfish (*Silurus glanis*).

The CPUE values gradually decreased from the shore to the center of the reservoir, both the mild slope ($p < 0.001$, deviance = 543.8, variance = 0.05) and the steep slope ($p < 0.01$, deviance = 554.7, variance = < 0.001), with little variance between the sites according to the model for both mild and steep slopes (Figure 9). When we compared the CPUE at the species level (Table 4), the perch (mild: $p < 0.001$, deviance = 296.6, variance < 0.001; and steep: $p < 0.001$, deviance = 257.5, variance < 0.001) and roach (mild: $p < 0.001$, deviance = 430.4, variance < 0.001; and steep: $p < 0.001$, deviance = 367.4, variance < 0.001) showed significant differences in both banks. Asp ($p = 0.013$, deviance = 156.6, variance < 0.001) and bream ($p < 0.015$, deviance = 179.9, variance = 9.215) showed a significant difference in the CPUE for the mild bank.

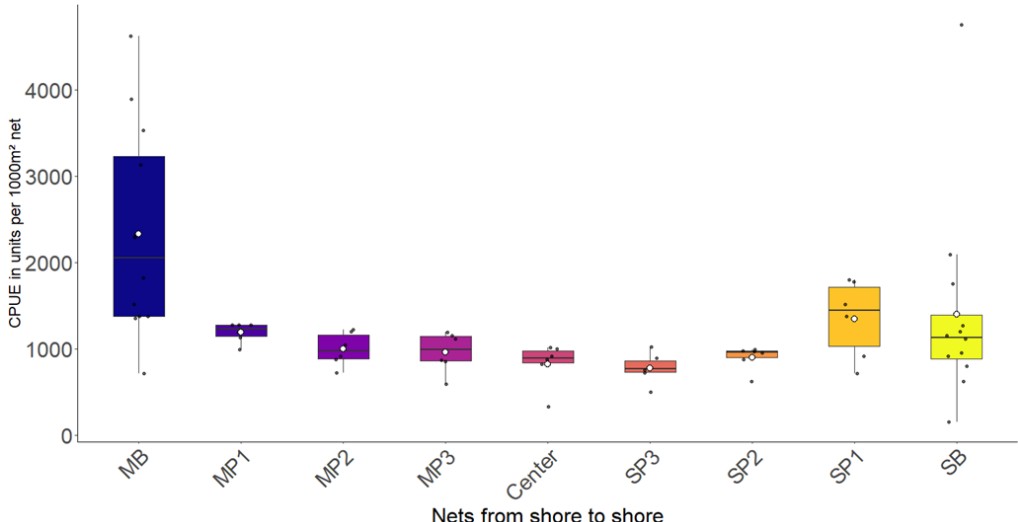

**Figure 9.** Total catch per unit effort (CPUE; individuals per 1000 m$^2$ of net) from 9 gillnet sites at different distances from the shore across the Řimov Reservoir. The boxplot represents the quartile value of the CPUE, the grey dots represent the mean of the individual locality, the thick middle line represents the median, and the white dot represents the arithmetic mean.

**Table 4.** Catch per unit of effort (inds. 1000 m$^{-2}$ of gillnets), standard errors (*se*), and the significance level (*p*) at various benthic and pelagic sites of Římov Reservoir. See Section 2 for detailed explanations of gillnet locations.

| Species | MB | se | MP1 | se | MP2 | se | MP3 | se | Center | se | p_Value |
|---|---|---|---|---|---|---|---|---|---|---|---|
| *Abramis brama* | 42.59 | 16.74 | 12.96 | 4.46 | 3.7 | 0 | 1.85 | 0 | 3.7 | 0 | 0.015 |
| *Alburnus alburnus* | 1220 | 312.11 | 1083 | 50.25 | 924.07 | 77.8 | 903.7 | 89.57 | 775.93 | 91.06 | ns |
| *Gymnocephalus cernua* | 112.96 | 25.87 | 0 | 0 | 0 | 0 | 0 | 0 | 0 | 0 | ns |
| *Leuciscus aspius* | 37.04 | 16.26 | 7.41 | 4.68 | 5.56 | 3.8 | 0 | 0 | 1.85 | 0 | 0.013 |
| *Perca fluviatilis* | 327.78 | 59.37 | 18.52 | 5.49 | 3.7 | 2.34 | 1.85 | 1.85 | 1.85 | 0 | 0.0001 |
| *Rutilus rutilus* | 561.11 | 68.45 | 62.96 | 19.6 | 55.56 | 10.34 | 48.15 | 9.37 | 38.89 | 11.74 | 0.0001 |
| *Sander lucioperca* | 24.07 | 6.95 | 3.7 | 2.34 | 3.7 | 2.34 | 3.7 | 2.34 | 3.7 | 0 | ns |
| *Scardinius erythrophthalmus* | 0 | 0 | 0 | 0 | 0 | 0 | 1.85 | 0 | 0 | 0 | ns |
| *Silurus glanis* | 1.85 | 1.85 | 1.85 | 0 | 0 | 0 | 0 | 0 | 0 | 0 | ns |
| **Species** | **SB** | **se** | **SP1** | **se** | **SP2** | **se** | **SP3** | **se** | **Center** | **se** | **p_Value** |
| *Abramis brama* | 18.52 | 9 | 1.85 | 0 | 3.7 | 2.34 | 16.6 | 12.75 | 3.7 | 0 | ns |
| *Alburnus alburnus* | 520.37 | 262.7 | 1200 | 166.27 | 837.04 | 64.26 | 705.56 | 58.36 | 775.93 | 91.06 | ns |
| *Gymnocephalus cernua* | 122.22 | 28.23 | 0 | 0 | 0 | 0 | 0 | 0 | 0 | 0 | ns |
| *Leuciscus aspius* | 22.22 | 7.74 | 18.52 | 6.2 | 7.41 | 4.68 | 5.56 | 3.8 | 1.85 | 0 | ns |
| *Perca fluviatilis* | 166.67 | 21.1 | 22.22 | 14.05 | 3.7 | 2.34 | 3.7 | 2.34 | 1.85 | 0 | 0.0001 |
| *Rutilus rutilus* | 529.63 | 107.93 | 96.3 | 24.29 | 44.44 | 11.83 | 40.74 | 6.83 | 38.89 | 11.74 | 0.0001 |
| *Sander lucioperca* | 7.41 | 4.18 | 1.85 | 0 | 1.85 | 0 | 3.7 | 0 | 3.7 | 3.7 | ns |
| *Scardinius erythrophthalmus* | 3.7 | 2.5 | 7.41 | 2.34 | 0 | 0 | 1.85 | 0 | 0 | 0 | ns |
| *Silurus glanis* | 7.41 | 3.16 | 0 | 0 | 0 | 0 | 0 | 0 | 0 | 0 | ns |

As for the BPUE, the values gradually decreased from the shore to the center of the reservoir, both on the mild slopes ($p < 0.001$, deviance = 312.0, variance = < 0.001) and on the steep slopes ($p < 0.001$, deviance = 285.3, variance = 0.05), with low variance among the localities according to the model for both mild and steep slopes (Figure 10). The perch (mild: $p < 0.001$, deviance = 158.5, variance < 0.001; and steep: $p < 0.001$, deviance = 143.9, variance < 0.01) and roach (mild: $p < 0.001$, deviance = 223.4, variance < 0.001; and steep: $p < 0.05$, deviance = 155.7, variance = 0.479) showed differences in distance for both banks, while bream ($p = 0.0026$, deviance = 97.0, variance = 2.557) showed differences just for the mild shore, and asp ($p = 0.043$, deviance = 117.1, variance = 0.28) for the steep shore (Table 5). Ruffe was caught only in the benthic gillnets.

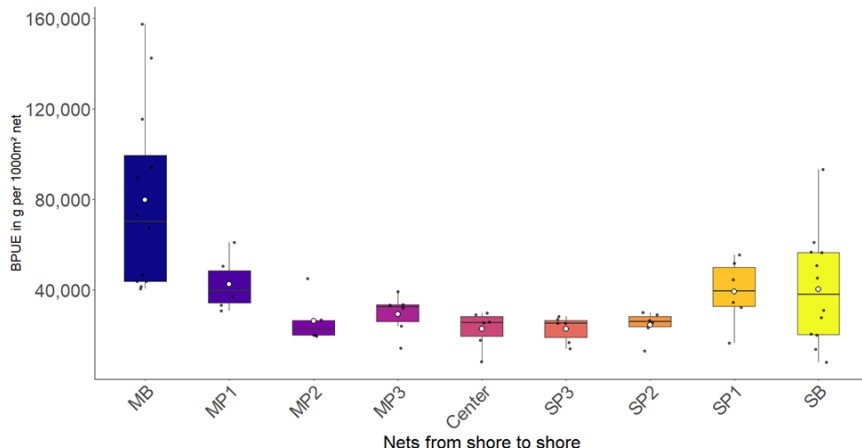

**Figure 10.** Total biomass per unit effort (BPUE; kg per 1000 m$^2$ of net) from nine gillnet sites at different distances from the shore across the Římov Reservoir. The boxplot represents the quartile value of the BPUE, the grey dots represent the mean of the individual locality, the thick middle line represents the median, and the white dot represents the arithmetic mean.

**Table 5.** Biomass catch per unit of effort (BPUE, g.1000 m$^{-2}$ of gillnets), standard errors (*se*), and the significance level (*p*) at various benthic and pelagic sites of Římov Reservoir. See Section 2 for detailed explanations of gillnet locations.

| Species | MB | se | MP1 | se | MP2 | se | MP3 | se | Center | se | p_value |
|---|---|---|---|---|---|---|---|---|---|---|---|
| *Abramis brama* | 6050 | 3132 | 3118 | 2085 | 37 | 0 | 70.44 | 0 | 133 | 0 | 0.0026 |
| *Alburnus alburnus* | 24,742 | 5686 | 24,826 | 1582 | 19,927 | 11,885 | 21,618 | 2746 | 17,156 | 2029 | ns |
| *Gymnocephalus cernua* | 645 | 145 | 0 | 0 | 0 | 0 | 0 | 0 | 0 | 0 | ns |
| *Leuciscus aspius* | 5986 | 2811 | 840 | 589 | 803 | 618 | 0 | 0 | 1065 | 0 | ns |
| *Perca fluviatilis* | 16,799 | 2945 | 3702 | 1119 | 608 | 420 | 230 | 0 | 249 | 0 | 0.0001 |
| *Rutilus rutilus* | 19,992 | 2805 | 7441 | 3326 | 4307 | 3116 | 4236 | 1831 | 2391 | 970.4 | 0.0001 |
| *Sander lucioperca* | 5084 | 1656 | 1235 | 784 | 574 | 571 | 2348 | 2068 | 1749 | 0 | ns |
| *Scardinius erythrophthalmus* | 0 | 0 | 0 | 0 | 0 | 0 | 972 | 0 | 0 | 0 | ns |
| *Silurus glanis* | 459 | 0 | 1435 | 0 | 0 | 0 | 0 | 0 | 0 | 0 | ns |
| **Species** | **SB** | *se* | **SP1** | *se* | **SP2** | *se* | **SP3** | *se* | **Center** | *se* | *p*_value |
| *Abramis brama* | 297 | 140 | 1653 | 0 | 143 | 98 | 568 | 364 | 132 | 0 | ns |
| *Alburnus alburnus* | 9610 | 4590 | 24,794 | 2798 | 19,455 | 1948 | 15,913 | 1347 | 17,156 | 2029 | ns |
| *Gymnocephalus cernua* | 774 | 138 | 0 | 0 | 0 | 0 | 0 | 0 | 0 | 0 | ns |
| *Leuciscus aspius* | 3598 | 2065 | 4690 | 2324 | 1475 | 976 | 787 | 690 | 1065 | 0 | 0.043 |
| *Perca fluviatilis* | 9545 | 2346 | 4199 | 2069 | 682 | 524 | 891 | 613 | 249 | 0 | 0.0001 |
| *Rutilus rutilus* | 11,999 | 3441 | 1928 | 737 | 1972 | 822 | 1692 | 1185 | 2391 | 970 | 0.05 |
| *Sander lucioperca* | 2816 | 2228 | 287 | 0 | 916 | 0 | 2093 | 0 | 1748 | 0 | ns |
| *Scardinius erythrophthalmus* | 1332 | 921 | 1733 | 937 | 0 | 0 | 913 | 0 | 0 | 0 | ns |
| *Silurus glanis* | 496 | 301 | 0 | 0 | 0 | 0 | 0 | 0 | 0 | 0 | ns |

Bleak superdominance was evident in all the pelagic samples (Figure 11). Only in the benthic samples on both sides of the lake did other species make up a larger proportion. When the biomass was expressed, the dominance of bleak persisted but was less evident (Figure 12). The species composition shows a gradual change from the shore to the open water, where the first pelagic net showed a species composition between the benthic and the pelagic habitat (still a conspicuous presence of roach, perch, bream, and asp). Ruffe is the best indicator of the benthic habitat, followed by the perch. Pikeperch, asp, catfish, and bream were caught in the open water, but their proportion was often lower than near shore. Rudd was not abundant, but also behaved like a eurytopic species, showing a homogeneous horizontal distribution. The distinct pattern of species distribution is reflected in a clear pattern of diversity indices (Figure 13). Species richness and diversity were always highest in the nearshore habitat and decreased towards the center of the reservoir. The Shannon index ($p = 0.0316$) and the number of species ($p = 0.0040$) showed a significant negative trend from the littoral to the pelagic. The size distribution also showed that the lowest mean lengths were found in the first nets on each side of the reservoir and the highest mean length was found in the center ($p < 0.001$, deviance = 56,291.8, variance = 0.0004, Table 6).

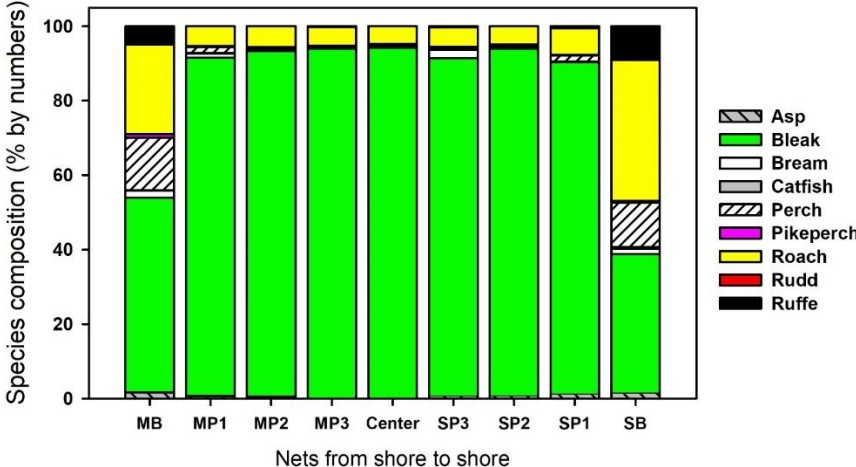

**Figure 11.** Fish species percentual numerical composition at different distances from the shore of Římov Reservoir.

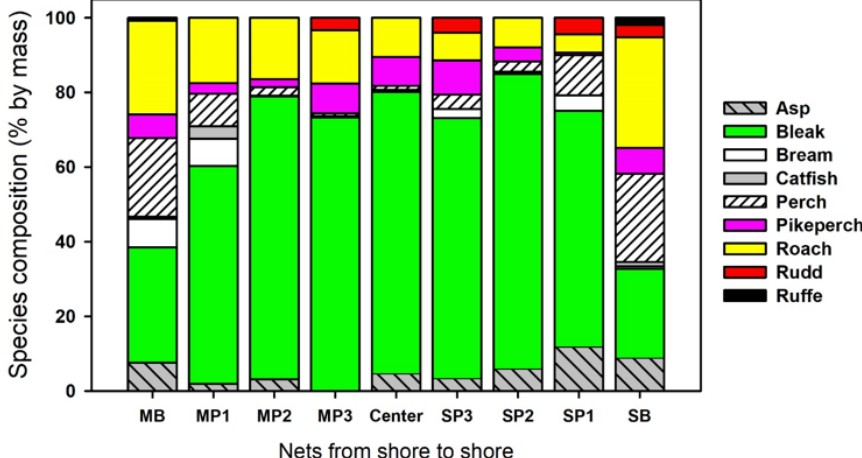

**Figure 12.** Fish species biomass composition at different distances from the shore of Římov Reservoir.

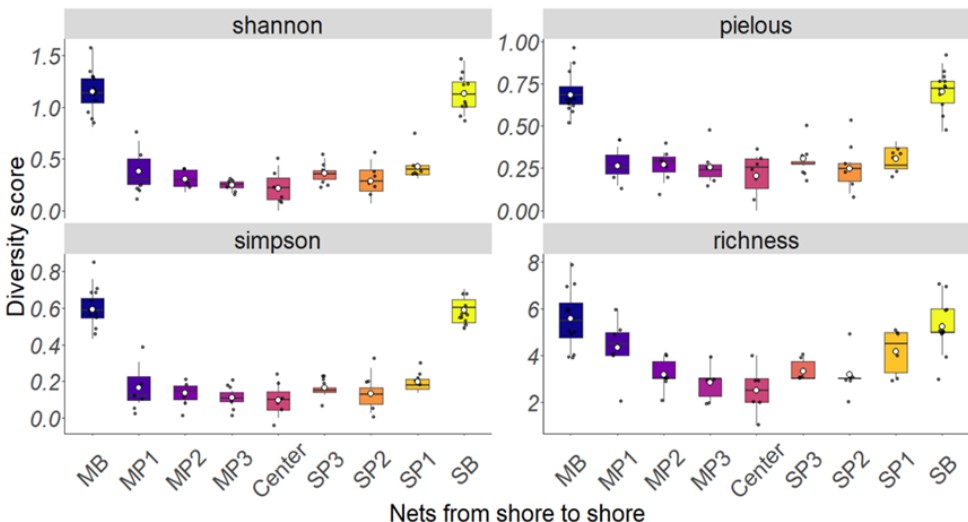

**Figure 13.** Fish diversity indices at different distances from the shore of Římov Reservoir.

**Table 6.** Average standard length (mm) and standard errors (*se*) of individual species at various benthic and pelagic sites of Římov Reservoir. See Section 2 for detailed explanations of gillnet locations.

| Species | MB | *se* | MP1 | *se* | MP2 | *se* | MP3 | *se* | Center | *se* |
|---|---|---|---|---|---|---|---|---|---|---|
| *Abramis brama* | 150.13 | *15.24* | 165.14 | *41.07* | 67.5 | *27.5* | 120 | *0* | 116 | *14* |
| *Alburnus alburnus* | 111.64 | *0.65* | 116.69 | *0.63* | 114.27 | *0.71* | 118.1 | *0.73* | 115.05 | *0.8* |
| *Gymnocephalus cernua* | 60.02 | *1.93* | - | - | - | - | - | - | - | - |
| *Leuciscus aspius* | 193.8 | *16.6* | 187.5 | *24.19* | 206.67 | *26.82* | - | - | 340 | *0* |
| *Perca fluviatilis* | 96 | *5.17* | 214.5 | *7.65* | 200 | *25* | 185 | *0* | 190 | *0* |
| *Rutilus rutilus* | 104.15 | *2.17* | 136.5 | *13.04* | 113.63 | *12.53* | 121.19 | *13.69* | 99.62 | *14.24* |
| *Sander lucioperca* | 241.08 | *17.53* | 297.5 | *7.5* | 168 | *122* | 334 | *116* | 332.5 | *27.5* |
| *Scardinius erythrophthalmus* | - | - | - | - | - | - | 265 | *0* | - | - |
| *Silurus glanis* | 340 | *0* | 480 | *0* | - | - | - | - | - | - |
| **Species** | **SB** | *se* | **SP1** | *se* | **SP2** | *se* | **SP3** | *se* | **Center** | *se* |
| *Abramis brama* | 88.8 | *3.75* | 340 | *0* | 119 | *14* | 110 | *8.85* | 116 | *14* |
| *Alburnus alburnus* | 108.79 | *0.89* | 112.24 | *0.67* | 116.79 | *0.79* | 115.07 | *0.93* | 115.05 | *0.8* |
| *Gymnocephalus cernua* | 63.85 | *1.35* | - | - | - | - | - | - | - | - |
| *Leuciscus aspius* | 187.83 | *23.18* | 228.1 | *28.02* | 221.25 | *36.08* | 193.33 | *43.43* | 340 | *0* |
| *Perca fluviatilis* | 108.23 | *6.85* | 196.58 | *18.77* | 202.5 | *42.5* | 227.5 | *27.5* | 190 | *0* |
| *Rutilus rutilus* | 88.93 | *1.92* | 85.98 | *4.16* | 95.58 | *10.94* | 98.73 | *10.73* | 99.62 | *14.24* |
| *Sander lucioperca* | 260 | *69.13* | 230 | *0* | 340 | *0* | 341.5 | *71.5* | 332.5 | *27.5* |
| *Scardinius erythrophthalmus* | 235 | *15* | 188.75 | *33.44* | - | - | 260 | *0* | - | - |
| *Silurus glanis* | 206.25 | *39.34* | - | - | - | - | - | - | - | - |

The mean densities of *Dapnia galeata* (the main food of non-predatory fish) were slightly higher at the littoral of the mild slope but were not significantly different from the other sampling stations along the transverse profile, with the exception of sites SP2 and SP1 (Figure 14). We also divided *D. galeata* into two size classes (small: body size ≤ 1 mm;

large: body size > 1 mm) and tested whether the densities of these size classes differed along the cross-section. Densities of small and large *Daphnia* were higher on average at the littoral of the mild slope littoral but were generally not significantly different from the other sampling stations, except for SP2 (small *Daphnia*), SP1, and SB (large *Daphnia*; Figure 15). The other two groups of zooplankton, other Cladocera, and Copepoda, were evenly distributed across the transverse profile (Figure 14).

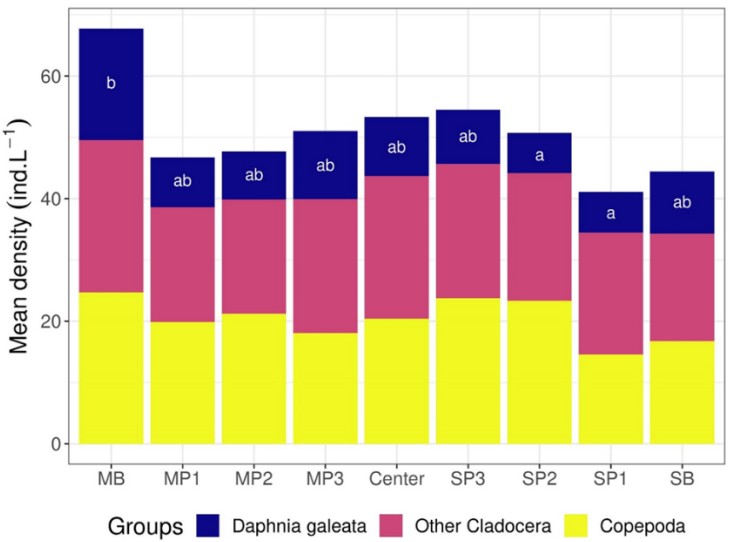

**Figure 14.** Mean density of three zooplankton groups (*Daphnia galeata*, other Cladocera, and Copepoda) at different distances from the shore of Římov Reservoir. Different letters indicate significant differences (*p* < 0.05) in *D. galeata* density between different distances from shore to shore. The densities of other Cladocera and Copepoda did not differ across the transverse profile (*p* > 0.05). Letters a and b denominate significant differences in *D. galeata* densities.

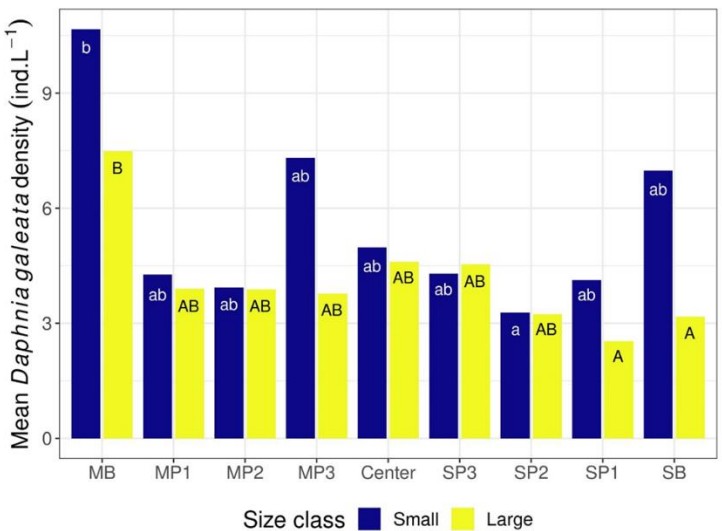

**Figure 15.** Mean density of small (≤1 mm) and large (>1 mm) *Daphnia galeata* at different distances from the shore of Římov Reservoir. Significant differences (*p* < 0.05) in the density of small *D. galeata* between different distances from shore to shore are indicated by different lowercase letters. Significant differences (*p* < 0.05) in the density of large *D. galeata* between different distances from shore to shore are indicated by different uppercase letters.

Benthic macroinvertebrates were generally more abundant on the gentle slopes (Figure 16). A significant difference between the mild and steep sites was found for the

Chironomidae ($p$ = 0.02, deviance = 101.6, variance < 0.001), Ephemeroptera ($p$ < 0.001, deviance = 161.7, variance = 0.060), and permanent fauna ($p$ < 0.001, deviance = 116.7, variance < 0.001) groups. For the difference in depth, only the permanent fauna was significantly less abundant in deeper water ($p$ < 0.001, deviance = 123.2, variance < 0.001).

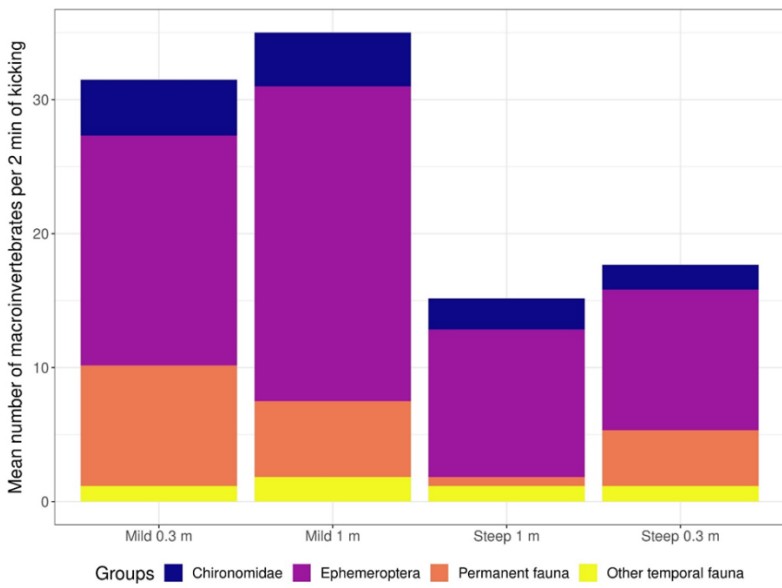

**Figure 16.** Mean number of benthic macroinvertebrates collected by kick-sampling at two depths on mild and steep slopes of Římov Reservoir.

## 4. Discussion

Our experiments have shown that the fish community changes very abruptly from the littoral to the pelagic in two different systems just near the benthic habitat. At the first pelagic point only 0.5 m above the bottom, the proportion of littoral species abruptly decreased. The pelagic habitat showed a homogeneous fish community composition, with a slight gradient corresponding to the distance from the littoral. This result supports previous assumptions that the definition of the benthic habitat only applies within a few meters of the bottom, and that the assumed height of the benthic habitat of 1.5 m above the bottom [31,36] may be accurate. The exception was the mild slope of Lake Most (site MP1), where the presence of abundant macrophytes created conditions that were very different from the pelagic habitat. The results also support previous assumptions that the pelagic habitat is the main volume even in medium-sized lakes, and that large volumes of open water must be considered if representative fish community values are to be obtained for the entire lake. Our results provide reassurance that the volume of the pelagic habitat is as large as estimated in previous studies [33,35] and that it is by far the most important habitat, even in relatively small waters [36].

The majority of species showed that they were benthic-bound, such as perch, ruffe, bream, pikeperch, asp, and roach (in some BPUE, also catfish). Bleak and rudd were determined to be typical eurytopic species. No exclusively pelagic species was found, which is consistent with the theory of Fernando and Holčík [24] about the scarcity of truly pelagic fish in young ecosystems. Consequently, the transition from the littoral to the pelagic community is mainly characterised by a sharp decline in the abundance and proportion of benthic species. This reflects the fact that the pelagic community is much simpler and less diverse, with fewer fish species willing to leave the safety of the littoral [24,26]. The results of this study showed that the fish community changes very quickly on the way from the bottom to the open water, and what we may call the pelagic community shapes most of the volume of lakes and reservoirs. This supports previous studies indicating that volume-weighted estimates provide much more realistic estimates for entire lakes than the global CEN CPUE [33].

The gradual decrease in fish abundance from the littoral to the pelagic zone in the middle of the lake was more evident in Most than in Římov. One reason for this difference could be the higher complexity of the habitat in the littoral of Most, due to the lower steepness and the high macrophyte density in the littoral zone, or the higher steepness in Římov. Littoral aquatic macrophytes are important components of habitat complexity and heterogeneity, as they dominate the nearshore zones of lakes and support diverse fish communities [48,49]. Macrophytes can influence fish habitat selection and ecological relationships such as predation and competition, which in turn affect the fish community structure. For example, predators may induce their prey to seek shelter in roots, leaves, and stems, which act as visual and physical barriers and provide protection from predators [27,50], while competition may induce fish individuals to seek new feeding grounds and reproduce [51]. Macrophyte habitats are considered nursery grounds for juvenile fish because they provide numerous sheltering opportunities, as smaller fish are more vulnerable to predators than larger fish [52,53]. The high macrophyte stems most likely caused very high fish catches at the first pelagic net at the mild slope of Most Lake.

However, even in the habitat without true aquatic macrophytes (Římov Reservoir), the CPUE, BPUE, and species diversity were mostly higher in the benthic habitats. This indicates that for many species at least the presence of some substrate is also important. The comparison between the benthic net catches shows that the mild slope "beach-type" habitats contained more fish than the steep slopes. Although the steep slopes may be more structured by rocks and tree remains [30,54], they are more open and clearly less safe for prey fish (see also [40]). The soft bottom substrate of mild slope shores is more favorable for benthic macroinvertebrates, and habitats with gentle slopes have also been found to have slightly higher densities of cladoceran *D. galeata*. According to previous studies carried out in the Římov Reservoir, cladoceran *D. galeata* is the main prey of the dominant non-predatory fish species [18,55,56]. Therefore, the reason for fish staying in the mild slope littoral of the Římov Reservoir could be both the protection from predators and feeding on *D. galeata* and the available benthic resources [30]. Other Cladocera (mostly represented by small species such as *Diaphanosoma brachyurum* and *Eubosmina coregoni*) and Copepoda were evenly distributed across the transverse profile of this reservoir and therefore did not appear to affect fish distribution. In general, the lowest average fish lengths were found in the littoral habitats, suggesting that juvenile fish feel more secure in the nearshore zone. This is in general agreement with the results of previous studies from other limnetic ecosystems [27]. While fish densities in the littoral mild slope habitats were considerably higher than in the open water, the CPUE and BPUE in the littoral habitats with the steep slopes were similar to those in the pelagic area. This may also be because in the steep slope habitats, the first pelagic net above the 3.5 m isobath was necessarily very close to the shore.

Our study only has a horizontal dimension. It deals with a layer of 0–3 m, which is normally the most populated by fish [8,11]. It was beyond our capabilities to extend the study to deeper habitats. However, the results from the 3 m depth are quite convincing, and we cannot expect the situation to change significantly in further layers. Fishes that require the substrate tend to stay close to it [57], while eurytopic fishes disperse without much regard to the benthic habitat.

## 5. Conclusions

Our experiment showed that the littoral zone was characterized by high numbers of fish, especially perch, and by the presence of smaller individuals. The catch of the pelagic nets was dominated by eurytopic fish—rudd and roach in Most and bleak in Římov. With the exception of one case where abundant macrophytes extended the structured habitat, the largest shift from the benthic to the pelagic community was observed only at the first pelagic gillnet at a bottom depth of 3.5 m. Open water catches were relatively consistent with small signs of a gradient towards the middle of the lake. The results indicate that the benthic gillnet catch is representative of a very limited area and volume, while most of

the volume is dominated by the pelagic community, the most important habitat even in relatively small waters. This has important consequences for the assessment of community parameters of the whole lake.

**Author Contributions:** Conceptualization, K.M., A.T.S., M.V., P.B., M.Č., T.J., J.M., J.P., M.Ř., M.Š., and J.K.; investigation and manuscript preparation and writing, K.M., A.T.S., M.V., D.B., P.B., M.Č., R.A.d.S., V.D., M.H., T.J., L.K., K.K., J.M., J.P., M.Ř., Z.S., M.Š., L.T., and J.K.; resources and funding acquisition, J.K., P.B., and J.P.; data curation, K.M., A.T.S., M.V., and P.B. All authors have read and agreed to the published version of the manuscript.

**Funding:** The study was supported by the European Union within ESIF in the frame of Operational Programme Research, Development, and Education (project no. CZ.02.1.01/0.0/0.0/16_025/0007417, Biomanipulation as a tool for improving water quality of dam reservoirs), by the project QK1920011, "Methodology of predatory fish quantification in drinking-water reservoirs to optimize the management of aquatic ecosystems", and by the Czech Science Foundation (project no. 20-18005S).

**Institutional Review Board Statement:** The study was conducted according to the guidelines of the Declaration of Helsinki, and approved by the Ethics Committee of the Biology Centre CAS vvi. And the Ministry of Environment of the Czech Republic Ref No. MZP/2019/630/16 dated 3 January 2019.

**Data Availability Statement:** The data are available in the database of the Biology Centre CAS and can be provided upon request.

**Acknowledgments:** Zdeněk Prachař helped with the preparation of figures. The help of Anjaly Menon, Kateřina Soukalová, and Tomáš Minařík during the field work is greatly appreciated.

**Conflicts of Interest:** The authors declare no conflict of interest. The funders had no role in the design of the study; in the collection, analyses, or interpretation of data; in the writing of the manuscript, or in the decision to publish the results.

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
