# Peer review of "Openness of Fish Habitat Matters: Lake Pelagic Fish Community Starts Very Close to the Shore"

_water, doi:10.3390/w13223291_

Round 1

Reviewer 1 Report

I believe that study is interesting, but I have the feeling that senior authors ought to check detailly the manuscript. There are a lot of minor mistakes, more of them very simple and ordinary. Please, take care with the format of the text. You can use the template of the journal to fix it.

I am not sure if this title is realistic, considering that research has been developed in lakes and not in rivers. The title should include the reference to lakes or lentic ecosystems almost.

Please, revise the legends of tables, all the abbreviations should be explained on the legend, or it should be a reference of them in the text. Legends should be independent texts.

Regarding results and discussion, it seems that these sections could be improved. The results about macroinvertebrates, zooplankton and Daphnia are poorly discussed.

In my opinion, section 4.1 (Future perspectives, what could be done better) should be eliminated. I think that these ideas are not related to the aim, results and conclusions of this study; this section is another thing that is not a part of this study.

Finally, I think that conclusions are not especially relevant. The presence of more fishes (and more diversity) on the shores and changes in pelagic areas are widely known.

Figure 3: please, include the capital letters (A, B, C) in the figure.

Line 6: Please, move ‘and’ before the last author

Line 35: fix ‘ecolog-ical’

Line 38: fix ‘organ-isms’

Line 48: fix ‘eco-systems’

Lines 62-64: scientific names in italics, please

Line 71: fix ‘re-flec-‘

Line 85: remove ‘which differ in terms of macrophyte presence, transparency, productivity and other variables’, explained in the next section.

Line 95: please, delete the bullet list; this section (Sampling sites) is not a list.

Line 140: what is the distance among gillnets? And how many nets are you used? This information should be included in this section.

Line 164-166: change ‘m2’ to superscript.

Line 236: delete the point.

Line 247: figure legend should be in the format like a figure legend.

Table 1 is the same as figure 4 (not just the same, but similar): I believe that figure 4 is unnecessary.

Figures 6, 7, 11 and 12. I think that the colours should be revised. In my opinion, there is not the best colour combination; it is not clear to see.

Line 446: please, delete the bullet list; this section (Sampling sites) is not a list.

Line 562: change ‘writng’ to ‘writing’.

Lines 573-577: please, delete this paragraph, or include a link to the data repository.

Author Response

I believe that study is interesting, but I have the feeling that senior authors ought to check detailly the manuscript. There are a lot of minor mistakes, more of them very simple and ordinary. Please, take care with the format of the text. You can use the template of the journal to fix it.

Thank you for thorough Review. We have checked the manuscript carefully and corrected a number of small errors. We apologize for them.

I am not sure if this title is realistic, considering that research has been developed in lakes and not in rivers. The title should include the reference to lakes or lentic ecosystems almost.

We have changed the title by adding a word lake so it is clear that our findings are relevant to lenitic ecosystems.

Please, revise the legends of tables, all the abbreviations should be explained on the legend, or it should be a reference of them in the text. Legends should be independent texts.

We have done complete revision of legends.

Regarding results and discussion, it seems that these sections could be improved. The results about macroinvertebrates, zooplankton and Daphnia are poorly discussed.

We improved the discussion in general and the results about macroinvertebrates, zooplankton and Daphnia in particular. We feel the discussion is now more clear.

In my opinion, section 4.1 (Future perspectives, what could be done better) should be eliminated. I think that these ideas are not related to the aim, results and conclusions of this study; this section is another thing that is not a part of this study.

We agreed and removed section 4.1 as suggested

Finally, I think that conclusions are not especially relevant. The presence of more fishes (and more diversity) on the shores and changes in pelagic areas are widely known.

Conclusions were shortened and focused on the main finding – very abrupt change from littoral to pelagial community.

Figure 3: please, include the capital letters (A, B, C) in the figure.

Done as suggested.

Line 6: Please, move ‘and’ before the last author

Done as suggested.

Line 35: fix ‘ecolog-ical’

Done as suggested.

Line 38: fix ‘organ-isms’

Done as suggested.

Line 48: fix ‘eco-systems’

Done as suggested.

Lines 62-64: scientific names in italics, please

Done as suggested.

Line 71: fix ‘re-flec-‘

Done as suggested.

Line 85: remove ‘which differ in terms of macrophyte presence, transparency, productivity and other variables’, explained in the next section.

Done as suggested.

Line 95: please, delete the bullet list; this section (Sampling sites) is not a list.

Done as suggested.

Line 140: what is the distance among gillnets? And how many nets are you used? This information should be included in this section.

Numbers of nets and the distances between them were added.

Line 164-166: change ‘m2’ to superscript.

Done as suggested.

Line 236: delete the point.

Done as suggested.

Line 247: figure legend should be in the format like a figure legend.

Done as suggested.

Table 1 is the same as figure 4 (not just the same, but similar): I believe that figure 4 is unnecessary.

The content of the table and figure differs. Tab. 1 gives CPUE of individual species while Figure 4 gives CPUE of whole fish community. If we remove Figure 4, the reader would be forced to reconstruct CPUE of whole fish community in his/her mind what is not convenient. For this reason we decided to leave Figure 4 in the paper.

Figures 6, 7, 11 and 12. I think that the colours should be revised. In my opinion, there is not the best colour combination; it is not clear to see.

We originally attempted to choose graph colours according to coloration of individual species. We agree that these colours were not so clear so we replaced the figures with much clearer colour scale.

Line 446: please, delete the bullet list; this section (Sampling sites) is not a list.

Done as suggested.

Line 562: change ‘writng’ to ‘writing’.

Done as suggested.

Lines 573-577: please, delete this paragraph, or include a link to the data repository.

Done as suggested.

Reviewer 2 Report

The paper I received for review presents very interesting data that will allow for a significant improvement in the accuracy of estimating fish community parameters using European standard gillnets. I highly appreciate the enormous amount of work that has been done to carry out such extensive research.

However, I have found some shortcomings that need to be corrected. They are listed below.

1.        Lines 161-166: for some reason instead of a description of the Most Lake sampling design, we have here a repetition of the text from lines 152-157. The lack of this description has caused some difficulty in tracking results from this lake and obviously needs to be added.

2.        Line 175: to be consistent with the caption of Figure 3, it should be “Figure 3B” instead of “Figure 3b”

3.        Lines 248 and 356: should be “CPUE” instead of “BPUE”

4.        Line 249: am I assuming correctly that in Figure 4 the presence of 3 grey dots for each gillnet location means that fishing in Lake Most took place on three dates? See comment #1

5.        Lines 339-342: there is something missing in this sentence to make it sound correct

6.   Lines 438-440: I propose to complete this sentence with the information that Permanent Fauna was less abundant in deeper water

7.        In the case of tables, in order to improve their readability, I suggest writing standard errors values in italics and/or parameter values in bold

Author Response

The paper I received for review presents very interesting data that will allow for a significant improvement in the accuracy of estimating fish community parameters using European standard gillnets. I highly appreciate the enormous amount of work that has been done to carry out such extensive research.

 Thank you for nice evaluation.

However, I have found some shortcomings that need to be corrected. They are listed below.

  1. Lines 161-166: for some reason instead of a description of the Most Lake sampling design, we have here a repetition of the text from lines 152-157. The lack of this description has caused some difficulty in tracking results from this lake and obviously needs to be added.

We apologize for this mishap originating from copping of earlier versions of text. Proper Most Lake sampling design was returned to its right place and repetition was deleted.

  1. Line 175: to be consistent with the caption of Figure 3, it should be “Figure 3B” instead of “Figure 3b”

Done as suggested.

  1. Lines 248 and 356: should be “CPUE” instead of “BPUE”

Done as suggested.

  1. Line 249: am I assuming correctly that in Figure 4 the presence of 3 grey dots for each gillnet location means that fishing in Lake Most took place on three dates? See comment #1

3 grey dots mean three replicates, three nets installed at the same distance from the shore at the same date. Explanation was added to the legend and also at “Most Lake sampling design”.

  1. Lines 339-342: there is something missing in this sentence to make it sound correct

Missing verb was missing and was added.

  1. Lines 438-440: I propose to complete this sentence with the information that Permanent Fauna was less abundant in deeper water

Done as suggested.

  1. In the case of tables, in order to improve their readability, I suggest writing standard errors values in italics and/or parameter values in bold

Done as suggested.

Round 2

Reviewer 1 Report

I am sorry, but in the current new version, I cannot revise correctly. I am sending you this version (the version that I can check) where there is red text and red strikethrough text. I have the feeling that this is not the new version, but another version. 

For example, I suggest changing the title and the authors reply that added the word Lake to clarify. However, in this last version, 'Lake pelagic' is crossed out, and 'Pelagic' is written in red. 

I am sure that this is an unimportant mistake. Please, send us the correct version.

Round 3

Reviewer 1 Report

All my suggestions have been considered. Congrats on the study.